# Matrix Composite Coatings Deposited on AISI 4715 Steel by Powder Plasma-Transferred Arc Welding. Part 3. Comparison of the Brittle Fracture Resistance of Wear-Resistant Composite Layers Surfaced Using the PPTAW Method

**DOI:** 10.3390/ma14206066

**Published:** 2021-10-14

**Authors:** Artur Czupryński, Marcin Żuk

**Affiliations:** Welding Department, Faculty of Mechanical Engineering, Silesian University of Technology, Konarskiego 18A, 44-100 Gliwice, Poland; rmt5@polsl.pl

**Keywords:** PPTAW, cladding, deposition, impact strength, brittle fracture strength, tungsten carbide, titanium carbide, titanium diboride, synthetic polycrystalline diamond

## Abstract

This article is the last of a series of publications included in the MDPI special edition entitled *“Innovative Technologies and Materials for the Production of Mechanical, Thermal and Corrosion Wear-Resistant Surface Layers and Coatings”*. Powder plasma-transferred arc welding (PPTAW) was used to surface metal matrix composite (MMC) layers using a mixture of cobalt (Co3) and nickel (Ni3) alloy powders. These powders contained different proportions and types of hard reinforcing phases in the form of ceramic carbides (TiC and WC-W_2_C), titanium diboride (TiB_2_), and of tungsten-coated synthetic polycrystalline diamond (PD-W). The resistance of the composite layers to cracking under the influence of dynamic loading was determined using Charpy hammer impact tests. The results showed that the various interactions between the ceramic particles and the metal matrix significantly affected the formation process and porosity of the composite surfacing welds on the AISI 4715 low-alloy structural steel substrate. They also affected the distribution and proportion of reinforcing-phase particles in the matrix. The size, shape, and type of the ceramic reinforcement particles and the surfacing weld density significantly impacted the brittleness of the padded MMC layer. The fracture toughness increased upon decreasing the particle size of the hard reinforcing phase in the nickel alloy matrix and upon increasing the composite density. The calculated mean critical stress intensity factor *K*_Ic_ of the steel samples with deposited layers of cobalt alloy reinforced with TiC and PD-W particles was 4.3 MPa⋅$m^{\frac{1}{2}}$ higher than that of the nickel alloy reinforced with TiC and WC-W_2_C particles.

## 1. Introduction

Composites of multi-component ceramics have been the subject of research and wider applications in materials engineering [1,2]. Among strengthening ceramics, WC, TiC, and TiB_2_ are some of the most popular materials because of their large Young’s modulus, good thermal stability, low density, and chemical compatibility with metals such as iron, nickel, cobalt, and titanium [3]. To improve the thermal conductivity and abrasive wear and erosion resistance of the working surface of drilling tools used in mining, Sue et al. [4] developed an innovative material suitable for surfacing with powder plasma-transferred arc welding (PPTAW) or laser metal deposition (LMD) techniques. The composite consisted of spherical fused tungsten carbide (SFTC) particles, with WC-W_2_C evenly distributed in the Ni-Si-B matrix. Compared with conventional hard layers padded with composite powder containing WC-W_2_C tungsten carbide particles and Ni-Cr-Si-B-Fe alloy, the developed material was characterized by a significantly higher thermal resistance and lower wear by abrasion and erosion. This improved the durability and increased the economic efficiency of the steel bits and polycrystalline diamond compact (PDC).

In recent years, research has also been conducted on composites using TiB_2_-TiC as a hard reinforcing phase, which are promising materials for use in the surfacing of parts that are resistant to abrasive wear and high temperatures [5,6]. According to Baoshuai et al. [7], the hardness and wear resistance of the laser plane was significantly improved due to the presence of TiB_2_ particles in an iron matrix. Similar observations were made by Tijo et al. (2018) [8], who investigated the mechanical properties of a composite coating on a Ti-6Al-4V alloy matrix containing TiC-TiB_2_ particles, which was obtained via in situ plating using the TIG method. The coating showed high strength and high abrasion resistance during pin-on-disc tests. A prospective ceramic metal matrix reinforcement in overlaid composite layers is polycrystalline synthetic diamond (PD). The previously presented results of research on the tribological properties of composite layers welded with the PPTAW method showed that compared with the wear-resistant AR 400 steel, the addition of polycrystalline diamond particles to the nickel matrix increased the wear resistance of the metal-mineral type by more than 11 times [9]. In the case of the cobalt warp, the increase was almost 140 times [10]. The properties of ceramic materials used to reinforce metal matrixes in surfaced composite layers are presented in Table 1.

The mechanical and tribological properties of the composite surfacing layers are particularly important for preventive protection and the regeneration of contact surfaces of the inserts of drilling tools. In recent years, drilling in the oil and gas mining sector has predominantly used cutter drill bits with polycrystalline synthetic diamond compact (PDC) blades. Among diamond tools, they reduce the working load of the drilling machine. In terms of construction, they are characterized by a steel or matrix body and segmented, ribbed, or winged arrangement of their blades (Figure 1).

The steel body of a drill bit is made of one piece of heat-treated alloy steel and reinforced with inserts on the outer peripheral surface made of cemented carbide or polycrystalline synthetic diamond. The matrix body of the bit is manufactured by infiltrating tungsten carbide particles, macrocrystalline WC, or fused-and-crushed WC-W_2_C, or a mixture of them, with a Cu-Ni-Zn-Mn alloy [4]. The steel body is more resistant to shock loading than the matrix body. The main disadvantage of the steel body is that it is quite susceptible to abrasive wear and erosion due to contact with the rock material to be mined. To protect the steel core of the rock cutting core from damage caused by tribological wear during operation, a coating is usually sprayed onto its surface or a layer more resistant to abrasion and erosion is deposited on its surface. The overarching goal of producing a durable steel body is to provide a hardfacing filler that has abrasion, erosion, and impact resistance equal to or better than that of the matrix body.

So far, the commonly used methods to protect the surface of the PDC steel body against wear include flame surfacing with powders based on the Ni-Cr-Si-B-Fe alloy matrix with the addition of irregular particles of angular fused-and-crushed (FTC) WC-W_2_C with a diameter of 10–160 μm. Large particles of spherical fused tungsten carbide (SFTC) WC-W_2_C with a diameter of 750–1200 μm have also been used. According to Badish et al. [13], these layers ensure the correct operation of the PDC drill for over 10 years. Examples of the chemical compositions of filler materials used for the hardfacing of tools working in the extraction, mining, and cement industries are presented in Table 2.

In recent years, the geological conditions of the exploitation of mineral, oil, and natural gas deposits have become more difficult. The filler materials and surfacing technologies currently used in most drilling applications do not sufficiently protect the steel body of the PDC drill bit; therefore, research was undertaken to develop a next-generation composite (MMC), dedicated to plasma surfacing that effectively protects the working surface of the PDC steel drill body against abrasive wear, erosion, high temperatures, and impact loading [9,10].

Powders based on cobalt and nickel belonging to Co3 and Ni3 alloys (in accordance with EN 147000) [19] with a hard strengthening phase containing, among others, WC, TiC, TiB_2_ particles, and of tungsten-coated synthetic polycrystalline diamond (PD-W). Powders based on these alloys often contain non-metallic elements such as B and Si. Higher contents of silicon and boron in the cladding powder resulted in more efficient melting of matrix components and improved the wetting of the reinforcing phase particles in composites. At the same time, when Si >1%, the plastic properties of the padded layer decrease, and with higher boron contents, the tendency to form austenite grains increases. The content of silicon and boron in the produced PPTAW hardfacing powders were optimized individually for each of the alloys depending on the type, shape, and size of particles included in the reinforcing components of the matrix metal. At the stage of preliminary preparation of the filler material, the content of silicon and boron in the metallic powders was changed from 0 to 2.5% in 0.5% incremental steps. Plasma arc melting of small volumes of composite powders allowed to select the filler material with the best weldability properties. In general, the silicon and boron contents were slightly lower than that of commercial hardfacing alloys. The use of spherical ceramic particles in the form of fused tungsten carbide WC-W_2_C and polycrystalline synthetic diamond determined the stable feeding of the filler material to the weld pool and the continuity of the surfacing process. Moreover, these particles in combination with a matrix of alloys of the Ni3 and Co3 groups had the effect of high hardness and stress reduction in the composite layer. Due to the larger surface of the carbide, there was a higher proportion of WC-W_2_C and TiC particles with irregular shapes in the hard reinforcing phase of the composite. This required a slightly higher content of fluxing components in the powder mixture, which enabled the production of powders for the plasma surfacing of steel body bits and tools with a “matrix-type armor” for drilling applications. In materials science, it is important to properly qualify materials for specific engineering applications. Usually, this is done based on the results of mechanical tests, in which material parameters such as Young’s modulus, proportionality limit, yield point, or temporary strength are determined; however, these are not the only mechanical properties used in engineering practice. The parameter defined within the framework of linear-elastic fracture mechanics is also taken into account, i.e., the stress intensity factor, *K*_I_, which is used to determine the fracture toughness of a material. It is generally accepted that for composite coatings, the critical value of the stress intensity factor determines its mechanical resistance to cracking under an impact load; thus, the evaluation of the *K*_Ic_ parameter value is very important for ceramic-reinforced composite coatings produced in via in situ surfacing processes, where the percentage content of the composition varies depending on technological conditions. Determination of the critical values of fracture toughness *K*_Ic_, crack tip opening displacement (CTOD), δ_Ic_, or J_Ic_ integrals according to the *I* crack growth model in accordance with the standards [20,21] is difficult and burdensome and requires appropriate laboratory equipment. These difficulties result from the methods used to make and prepare test samples and the need to maintain appropriate test conditions and procedures [22]. For these reasons, the relationship between the results of standard tests of the mechanical properties of the material and the critical values of the fracture toughness parameters is constantly sought. Many research results on the fracture toughness of composite coatings are based on the indentation method and the formula proposed by Evans and Wilshaw [23]. Several researchers have used this method to evaluate the fracture toughness of a brittle composite coating developed using various methods and compared the results with the fracture toughness measured by standard methods [24,25]. These are most often the dependencies connecting the results of the work of impact breaking of samples with a Charpy V–KV sharp notch (expressed in Joule units) with the critical values *K*_Ic_ and *δ*_Ic_. There are also papers that discuss other methods for determining these values, e.g., determining the *K*_Ic_ value for plasma-sprayed coatings based on measuring the length of cracks around the impression formed during hardness measurements using the Vickers method [26]. Such alternative approaches for determining the fracture toughness of a material are generally applicable to industrial conditions. This article presents the analytical evaluation of the resistance to brittle fracture under a dynamic load of abrasion-resistant metal matrix composite layers for protection against wear of working surfaces of drilling tools used in the oil and natural gas mining sector. There is no information related to tests concerning the brittle fracture resistance of the MMC layers reinforced with metal-diamond composite, obtained using the powder plasma transferred arc welding or the laser metal deposition methods. The analysis of related reference publications and the results of personal research led to the conclusion that it is possible to obtain a ceramic reinforcement-metal matrix composite surface layer characterized by microstructure and brittle fracture resistance similar to the classic metal surfacing weld. The original article achievement is, by selecting the right research methodology and analytical tools, obtaining information about the shock load resistance of innovative composite layers.

## 2. Materials and Methods

For PPTAW, four different powders with a proprietary chemical composition were used—cobalt Co3 and nickel Ni3 matrix composites (chemical compositions in accordance with EN 147,000 [19]) containing super hard phases in the form of ceramic particles with crushed sharp-edged TiC, spherical fused tungsten carbide particles (SFTC) WC-W_2_C, fused tungsten carbide (FTC) WC-W_2_C, fine particles of titanium diboride TiB_2_, and spherical particles of polycrystalline synthetic diamond with a tungsten coating PD-W (Harmony Industry Diamond, Zhengzhou, China). The chemical composition of the hardfacing powders is given in Table 3. 

A SEM image of the micromorphology of the powder components used for PPTAW cladding is shown in Figure 2. The density of the powders was tested by the volumetric pycnometric method using an AccuPyc II 1340 density analyzer (Micromeritics Instrument Corporation, Norcross, GA, USA) with a measurement accuracy of 0.03%. The volume of the samples was determined as part of a previously marked measuring chamber that was not filled with gas (helium pressure 134,447.77 Pa). The obtained values of five measurements for each type of powder indicate the repeatability of its density. The powders were dried before use in an S 60/03 chamber dryer (LAC, Židlochovice, Czech Republic) in a 6-h cycle at 150 °C.

The particle sizes of the powders constituting the additional material for cladding were measured by laser diffraction using an Analysette 22 MicroTec plus laser particle size meter (Fritsch GmbH, Idar-Oberstein, Germany) equipped with two semiconductor laser sources: green (λ = 532 nm, 7 mW) and infrared (λ = 940 nm, 9 mW). Measurements were made using a wet dispersion unit with an ultrasonic exciter.

The surfacing tests were carried out on an automated welding station equipped with a Eutronic GAP 2501 DC power source, EP2 powder feeder, and E52 universal plasma torch (Castolin Eutectic, Gliwice, Poland), as shown in Figure 3. Single-pass padding was deposited on samples with dimensions of 75 × 10 × 8 mm, made of low-alloy AISI 4715 structural steel (Table 4). Surfacing was performed with constant technological parameters, which were the same for each type of powder (Table 5).

Fracture toughness tests of the parent material and steel samples with wear-resistant composite surfacing welds were carried out in accordance with ISO 148 [28] using a standard Charpy SUNPOC JB-300B impact hammer (Sunpoc, Guiyang City, China) with a hammer with a rounding radius in an impact point of 2 mm.

The apparent density, porosity, and water absorption of the composite layers were determined by Archimedes’ method in accordance with the ASTM D792-00 standard [29]. An AS 220.R2 Plus analytical balance (Radwag, Radom, Poland) with a reading accuracy of ±0.1 mg was used to measure the mass. The analytical balance was equipped with a set KIT 85 for determining the density of solids using the hydrostatic weighing method.

The metallographic examination of the structures of the wear-resistant surfacing layers and the parent material was carried out in accordance with ISO 17639 [30]. Metallographic specimens were prepared in a standard manner and were collected perpendicularly and parallel to the cladding direction. The polished specimens were etched in a mixture of concentrated hydrochloric acid and concentrated nitric acid in a volume ratio of 3:1. The etching time was selected experimentally, individually for each of the layer materials. Observation and registration of microstructure images were performed with an Olympus GX 71 inverted metallographic microscope (Olympus Corporation, Tokyo, Japan). The surface topography of impact fractures was studied using a Zeiss Supra 25 system (Carl Zeiss AG, Oberkochen, Germany) using the secondary electron (SE) detector, with an accelerating voltage of 20 kV.

## 3. Results

### 3.1. Particle Size Distribution in Powders

The particle size of the powders of the additional material for surfacing was measured in distilled water. Five measurements were made for each of the tested powders. Each test powder was ultrasonically dispersed for 5 s before and during the test. The overall measurement range was 0.08–2000 µm. The obtained results are presented in the form of particle size distributions (Figure 4), and the selected statistical values are collected in Table 6.

The test results showed that the cobalt alloy powder with a hard reinforcing phase in the form of particles of crushed titanium carbide TiC and spherical polycrystalline synthetic diamond sintered PD-W (C1 filler material) had the highest mean particle size (152 µm). On the other hand, the C4 filler material—a powder based on a nickel alloy matrix with a hard reinforcing phase in the form of particles made of titanium diboride TiB_2_ and spherical tungsten carbide WC-W_2_C—was characterized by the smallest average particle size, amounting to 82 µm.

### 3.2. Density and Porosity of Composite Surfacing Welds

Based on the average of three mass measurements of test samples taken from the weld metal of each surfacing weld, calculations were performed to characterize the physical properties of composite layers, i.e., apparent density, open porosity, and water absorption. Samples cut from the welds were dried in a laboratory dryer at 110 ± 5 °C to a constant mass and then cooled in a desiccator. After determining the dry mass of the tested materials, the samples were deaerated in a vacuum device, and then saturated with liquid. The apparent mass of the sample immersed in the liquid and the mass of the sample saturated with liquid and weighed in the air were determined. The results of measurements and calculations are presented in Table 7. An example view of the longitudinal section of the macrostructure of selected surfacing welds is shown in Figure 5.

The results of density measurements and calculations of the porosity of composite layers deposited with the PPTAW method showed that the weld metal obtained from powder based on a nickel alloy matrix with a hard reinforcing phase in the form of particles made of titanium diboride TiB_2_ and spherical tungsten carbide WC-W_2_C (C4 filler material) has the highest density (9.6 g/cm^3^) and the lowest porosity (2.7%). The lowest density (5.78 g/cm^3^) was found for the layer obtained from cobalt alloy powder with a hard reinforcing phase in the form of particles of crushed titanium carbide TiC and spherical polycrystalline synthetic diamond sintered PD-W (C1 filler material), and the highest porosity (11.2%) for the weld metal of the cladding weld made from nickel alloy powder with a hard reinforcing phase in the form of particles of WC-W_2_C spherical and broken tungsten carbide and PD-W spherical polycrystalline synthetic diamond sinter (C3 filler material).

### 3.3. Assessment of the Microstructure of Composite Surfacing Welds

The results of microscopic metallographic observations were used to determine the structure, type, distribution, and dimensions of the reinforcement phase and matrix of the composite wear-resistant layers in the near-surface, middle, and transitional zones of the surfacing weld. Light microscopy images of the structure were taken at a magnification of 200× (Figure 6). The microstructure and the results of the qualitative surface analysis in the polycrystalline synthetic diamond are shown in Figure 7 and Figure 8.

The tests involving the use of light microscopy revealed that the microstructure of the layer padded with a filler material C1 was dendritic, multidirectional, and contained numerous inclusions of ceramic particles of titanium carbide (TiC) as well as single particles of the synthetic metal-diamond composite (Figure 6a). The microstructure of the layer padded with a filler material C2 was composed of spherical particles of primary tungsten carbides and particles of crushed titanium carbide TiC in the nickel alloy matrix (Figure 6b). In turn in the microstructure of the layer padded with a filler material C3, morphologically diverse intermetallic phases consisting predominantly of spherical and crushed particles of primary tungsten carbide as well as smaller amounts of complex secondary carbides on the carbide–matrix boundary (Figure 6c) [9]. The secondary carbides were responsible for the diffusive bond of the primary carbides with the matrix. According to Bober et al. [3], the above-named mechanism of the bonding of carbides with the matrix should ensure their stable deposition. The partial melting of the primary carbides led to the partial saturation of the matrix with tungsten and carbon. In the structure of the padding weld made with C4 filler material, only different size particles of primary spherical tungsten carbide in the nickel alloy matrix were observed. The microstructure of alloy structural steel AISI 4715 is ferrite and lamellar perlite (Figure 6e).

### 3.4. Critical Parameters of the Material’s Resistance to Cracking

#### 3.4.1. Correlation Relationships Determining the Critical Parameters of Fracture Mechanics

Based on the work of the impact breaking *KV* (J) of the “Charpy V” type samples, the fracture toughness *K*_Ic_ (MPa⋅$m^{\frac{1}{2}}$) can be determined from the following relations [31]:(1)$$K_{\mathrm{Ic}}=\sqrt{0.00022\cdot E\cdot {\left(\mathrm{KV}\right)}^{\frac{3}{2}}}$$
(2)$$K_{\mathrm{Ic}}=\sqrt{0.00137\cdot E\cdot \left(\mathrm{KV}\right)}$$
(3)$$K_{\mathrm{Ic}}=14.5\cdot \sqrt{\left(\mathrm{KV}\right)}$$
(4)$$K_{\mathrm{Ic}}=0.53\cdot \left(\mathrm{KV}\right)+57.9$$
where, in the absence of material data concerning the value of Young’s modulus (*E*), Formula (3) is most often used.

On the other hand, the critical value of the fracture front opening, *CTOD* (units: mm) can be determined by Equation (5), as shown in [32]:(5)$$CTOD=\delta_{\mathrm{Ic}}=0.0024\cdot KV$$

#### 3.4.2. Test Results and Calculations of the Critical Values of K_Ic_ and CTOD (δ_Ic_)

The impact tests were performed to determine the brittleness threshold and the nature of the degradation of the AISI 4715 steel parent material and steel samples with wear-resistant composite surfacing welds. The work of the impact breaking *KV* was determined using a Charpy pendulum hammer with an initial energy of 300 J. The test was performed at 23 °C on standardized samples with a V-shaped notch with an angle of 45^o^, a depth of 2 mm, and a bottom rounding radius of 0.25 mm. A notch was cut from the underside of the sample on the side opposite to the surfacing weld (Figure 9). For each type of material, one test set was prepared that consisted of three samples with dimensions of 10 × 10 × 55 mm (full-sized sample). During the tests, the sample was adhered to the supports, the hammer impact was centered, the notch axis was in the plane of the hammer’s motion, and the notch was directed so that the hammer hit the surfacing weld during the test. The averages of test results and calculated *K*_Ic_ and *CTOD* values are summarized in Table 8. Moreover, the test results were supplemented with morphology and chemical composition analyses of fractured micro-areas recorded with a stereoscopic microscope (Figure 10) and a scanning electron microscope (Figure 11).

The SEM images of fracture surfaces of all samples with composite layers (Figure 11a–d) showed an obvious brittle fracture mechanism. Also, in the base material (Figure 11e) no specimen exhibited the presence of ductile fracture areas with the formation of typical dimples, brittle intergranular fracture dominated.

## 4. Discussion

In the first publication of the series, we reported the structural and tribological properties of composite layers padded using PPTAW that contained Co-Cr-W-Mo alloys (C1 filler material) and Ni-Cr-B-Si (C2, C3, and C4 filler materials). These filler materials contained various combinations and volume fractions of hard ceramic particles (TiC, WC-W_2_C, TiB_2,_ and PD-W) [9,10]. This earlier research showed that the surface plasma-welded layers made of these alloys displayed metal-mineral abrasion resistance, which made them suitable for the contact surfaces of the drilling tool inserts used in the oil and natural gas sectors. The current research was carried out to assess the resistance of the above-mentioned composite layers to impact loading.

### 4.1. Influence of Powder Particle Size on the Cracking of Surfaced Composite Layers under an Impact Load

Figure 2 shows the morphology of the components of the powders used for hardfacing the tested composite layers with the PPTAW method. The powders, apart from easily fusible matrix metal particles (Co3 and Ni3 alloys), whose diameter did not exceed 50 µm, also contained flame-retardant and super-hard phases in the form of ceramic particles from crushed, sharp-edged TiC (about 250 µm), spherical fused tungsten carbide WC-W_2_C (about 160 µm), broken tungsten carbide WC-W_2_C (about 80 µm), fine-grained titanium diboride TiB_2_ (about 40 µm), and a spherical polycrystalline synthetic diamond with a tungsten coating (about 60 µm). Measurements of the particle sizes of powders (Table 6) in relation to the relevant calculations of the critical stress values *K*_Ic_ and the fracture front *CTOD* (Table 8) showed that the particle size of the hard ceramic phase was closely related to the fracture toughness of the MMC layers (Figure 12).

The micrographs showed that in the composite layers on the Ni3 alloy matrix (C2, C3, C4 filler materials) large particles of sharply crushed TiC (Figure 6b and Figure 11b) cracked more easily than spherical particles of WC-W_2_C or PD-W (Figure 6c and Figure 11d). Larger TiC particles were more susceptible to structural defects than smaller ones [33]. The larger the particle size of the hard ceramic phase, the larger the interfacial surface (matrix–ceramic particles) and the greater the stress transferred from the interfacial surfaces to the particles. Large ceramic particles fixed in a relatively plastic matrix showed less tendency to plastic deformation of the composite layer.

It was noticed that during crystallization of the liquid metal in the weld pool (C1 filler material), some TiC particles cracked due to tensile stress concentration in carbide defects (Figure 6a). The relatively low Young’s modulus (Table 1) and the resulting reduction in the plasticity of TiC particles by strong intermolecular bonds indicates catastrophic crack propagation in the material. This thesis is confirmed by the research presented in refs. [34,35], which investigated the influence of tensile stress on the fracture of TiC particles during laser alloying of steel with cobalt alloys at different laser treatment parameters. The literature shows that the fracture toughness of TiC is 4–5.5 MPa·$m^{\frac{1}{2}}$ [36].

Based on SEM observations of fractures in the composite layers containing fused spherical WC-W_2_C particles (Figure 11d), it was found that the initiation of a single microcrack occurred inside the WC-W_2_C particles. The high temperature gradient resulting from the rapid heating and cooling of the composite layer welded with the PPTAW method, along with the 1/3 lower thermal expansion coefficient of the WC-W_2_C particle (Table 1) lower than the matrix metal, led to the generation of thermal stress in the ceramic particles. When the value of the thermal stress exceeded the yield point of the polycrystalline WC-W_2_C particle, microcracks formed on the surface of grain boundaries, spread, and then penetrated the particles. After exceeding the critical dimension, the microcrack propagated further from the WC-W_2_C particle into the composite matrix, easily developing along the fracture-prone and brittle dendritic and interdendritic eutectic phases. The number of brittle eutectic phases strongly influenced crack propagation. According to Zhou et al. [37] and Wang et al. [38], the thermal stress caused by a high temperature gradient and different coefficients of thermal expansion between the ceramic particle and matrix cause cracking of WC-W_2_C particles (Figure 6b,d). Often, cracks in WC-W_2_C particles and matrix continued to spread, merging to form macrocracks, as shown in Figure 11b. Xu et al. [39] also observed cracks of WC particles in the composite layer on the matrix of a nickel alloy padded with the LMD method. Researchers proposed that the critical value of residual stress was determined mainly by microstructural and thermal stresses and was the main cause of MMC fracture. Such microcracks, under the influence of dynamic loading, can initiate brittle cracks. Surface topography tests on the impact fractures of all composite layers and the base material (Figure 11c) showed a brittle fracture mechanism.

It was also found that WC-W_2_C melted at a much higher temperature during surfacing than the Ni-Cr-B-Si matrix material (C2, C3, C4 filler materials), as shown in Figure 6b–d. Depending on the amount of heat supplied to the material and the shape and size of the tungsten carbide particles, a small number of particles of the hard matrix strengthening phase dissolved during surfacing, which was also observed by Badish and Kirchgaßner [13]. A similar situation occurred in the case of synthetic polycrystalline diamond particles covered by a tungsten coating (PD-W), where, after partial melting of the coating, the tungsten was transferred to the matrix solution (Figure 8). The same observation was made by Telasang et al. [40], who investigated the microstructure and mechanical properties of laser-powder deposited layers using the direct metal deposition (DMD) technique on an AISI H13 tool steel substrate. As a consequence, during solidification, the tungsten and carbon pads entered the matrix solution. In general, the dissolution of tungsten and carbon strengthened the matrix and increased its abrasion resistance [9,10,41], but also reduced the fracture toughness and overall strength of the surfacing layers. According to Yan et al. [42], particle fracture is usually related to the highest principal stress, which increases upon increasing the particle size. As a result, small ceramic particles whose greatest principal stress was negligible were more difficult to fracture. In addition, the ceramic inclusions tended to break more easily at large aggregates where there was a high stress concentration; however, no cracks were observed in the particles of the polycrystalline synthetic diamond PD-W. The formation of an intermediate layer (~25 µm) between the polycrystalline synthetic diamond particle and the cobalt or nickel matrix (Figure 8), by melting the tungsten coating improved the stress transfer from the matrix to particles.

### 4.2. Influence of Density and Porosity on the Cracking of Surfaced Composite Layers under an Impact Load

Fracture toughness is a complex function of not only the physical properties of the matrix, or the type, proportion, and size of the hard phase reinforcing phase particles, but also the state of inter-grain and interfacial boundaries, as well as the stress in non-metallic inclusions and their surroundings. According to Bućko et al. [43], another important parameter is the density of a composite—often determined by its porosity. Figure 13 shows the joint effect of the matrix and ceramic inclusions, expressed by the specific density of the composite surfaced layer on the *K*_Ic_ value.

The lowest fracture toughness was obtained by the base material (AISI 4715 steel) without a surfacing layer, for which the calculated average *K*_Ic_ value did not exceed 48 MPa⋅$m^{\frac{1}{2}}$. In the case of composite layers surfaced by plasma with powders on a Ni3 alloy matrix, the highest value of the stress intensity factor *K*_Ic_ = 72.1 MPa⋅$m^{\frac{1}{2}}$ was recorded for the layer containing spherical particles of fused WC-W_2_C and fine particles of TiB_2_ (C4 filler material). Moreover, in the case of nickel-based filler materials (C2, C3, C4 filler materials), a linear relationship was observed between the density and fracture toughness of the surfaced composite layers. The coefficient of determination *R*^2^ = 0.9029 indicated that the presented regression equation is very useful for predicting the value of *K*_Ic_ using the specific density ρ. The high Pearson’s linear correlation coefficient *r* = 0.9933 suggests its significant influence on cracking. A large standard deviation σ for the calculated *K*_Ic_ values obtained at higher densities and decreasing upon decreasing the density, indicates that this material property is an important parameter that controls the fracture toughness of composites. The decrease in density decreased the fracture toughness. This has been confirmed by Rabin et al. [44] and Grabowy et al. [45]. It should be remembered that the graph shown in Figure 13 is merely an illustration of the qualitative tendency and not the quantitative dependence of the presented values. This is because the data come from measurements made on samples differing in both the type and size of the strengthening phase. On the other hand, the composite layer made of a powder on the matrix of Co3 alloy (C1 filler material) showed an average fracture toughness (*K*_Ic_ = 63 MPa⋅$m^{\frac{1}{2}}$) with a comparatively low density of the composite material (ρ = 5.78 g/cm^3^). The relatively good fracture toughness of this layer was attributed to the strong interfacial bonding of the polycrystalline synthetic diamond (PD-W) with the cobalt alloy matrix (Figure 7a). The density of the padded layer depended on the type and volume fraction of the hard matrix reinforcing phase and also on the total porosity of the composite layer. According to Bober et al. [12], when hard phase particles with a high melting point are well-wetted by the liquid metal matrix, correct surfacing welds are formed, and the usage degree of the hard reinforcing phase is high. On the other hand, when the wettability is insufficient, the ceramic particles are displaced from the liquid weld pool, and the degree of recovery of the strengthening particles is low. This usually leads to a series of welding imperfections, especially gas bubbles, external pores, and cracks in the padding layer. In addition to ensuring good wettability of the composite components, the density difference between the hard reinforcing phase and the matrix greatly influences the porosity of the surfacing welds. A significant difference in the specific mass of the matrix metal and the strengthening phase can lead to the uneven distribution or agglomeration of the strengthening particles in the surfacing weld metal. According to Gawdzińska et al. [46], local densification of ceramic particles may result in pores caused by insufficient saturation of capillary spaces between the reinforcement phase and liquid matrix metal. The test results of composite surfacing welds made of powder on a Ni3 alloy matrix do not unequivocally confirm the thesis that a reduction in fracture toughness is tantamount to an increase in the porosity of the layer (Figure 14).

Even when the total porosity *P*_c_ rises above 6%, other factors were responsible for the fracture toughness; therefore, it is possible to obtain composites that differ significantly in their *K*_Ic_ with a comparatively high total porosity (*P*_c_ > 6%). For the same reasons, an increase in *K*_Ic_ was observed for filler material C3. The analysis of Figure 14 does not allow us to conclude that with the same chemical composition of the matrix, the greatest impact on the critical value of *K*_Ic_ can be expected in the case of a high porosity *P*_c_. The low value of the correlation coefficient (−0.0826) does not show a rectilinear relationship between *K*_Ic_ and *P*_c_, which suggests that other factors have a more significant influence on the cracking mechanism, e.g., the type and volume fraction of hard-phase particles reinforcing the composite. The composite layer padded with powder on the matrix of the Co3 alloy (C1 filler material) with a total porosity *P*_c_ = 5.9% showed a crack resistance (*K*_Ic_) of 63 MPa⋅$m^{\frac{1}{2}}$.

## 5. Conclusions

Based on experimental tests and analytical calculations of the critical values determining the fracture toughness of the innovative metal matrix composite layers, it can be concluded that:(1)In the case of composite layers based on a nickel alloy matrix, the critical value of the stress intensity factor *K*_Ic_ decreased linearly upon increasing the powder particle size and increased linearly upon increasing the density of the composite layer.(2)Taking into account the comparable size of the powder particles as well as the density and porosity of the composite material, it was shown that the composite layer based on the cobalt alloy matrix (C1 filler material) had a slightly higher fracture toughness than the composite layer based on the nickel alloy matrix (C2 filler material). The critical value of the stress intensity factor *K*_Ic_ for the composite layer on the Co3 alloy matrix was 4.3 MPa⋅$m^{\frac{1}{2}}$ higher.(3)The nucleation of cracks in composite layers most often occurred inside the particles of the hard strengthening phases (TiC and WC-W_2_C). They propagated in the matrix along the very brittle dendritic and interdendritic eutectic phases of the composite microstructure, which were susceptible to cracking. Spherical particles made of polycrystalline synthetic diamond PD-W covered with a tungsten coating showed high thermal stability and no susceptibility to cracking during surfacing using the PPTAW method.(4)Thermal stresses occurring during plasma surfacing and the different thermal expansion coefficients of the ceramic reinforcement particles (TIC and WC-W_2_C) and the metallic composite matrix may have been the cause of crack initiation and propagation in the composite layers.

## 6. Patents

The procedure for granting a patent (No. P435997) was initiated before the Patent Office of the Republic of Poland.

## Figures and Tables

**Figure 1 materials-14-06066-f001:**
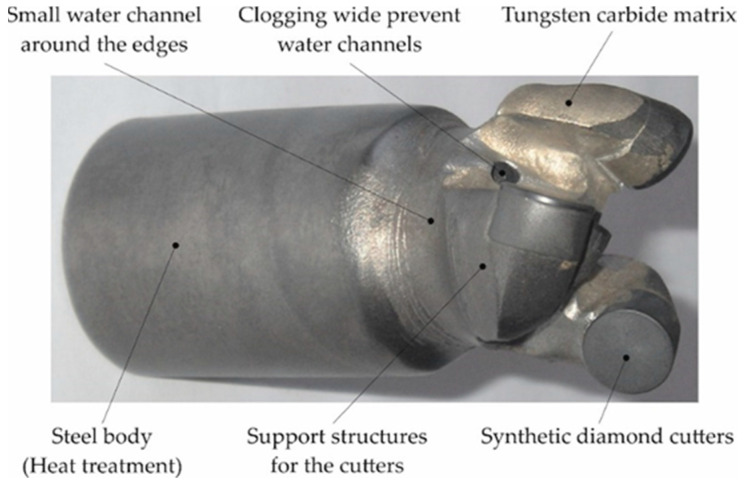
PDC small-diameter bit used in underground drilling.

**Figure 2 materials-14-06066-f002:**
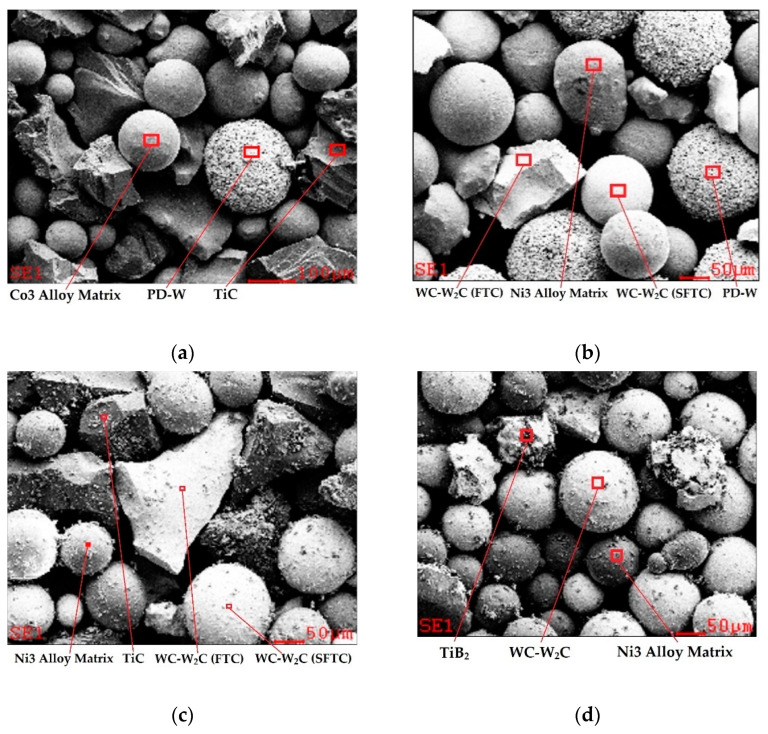
SEM image of the micromorphology of the powder components used for PPTAW cladding: (**a**) C1 filler material; (**b**) C2 filler material; (**c**) C3 filler material; (**d**) C4 filler material (Table 3).

**Figure 3 materials-14-06066-f003:**
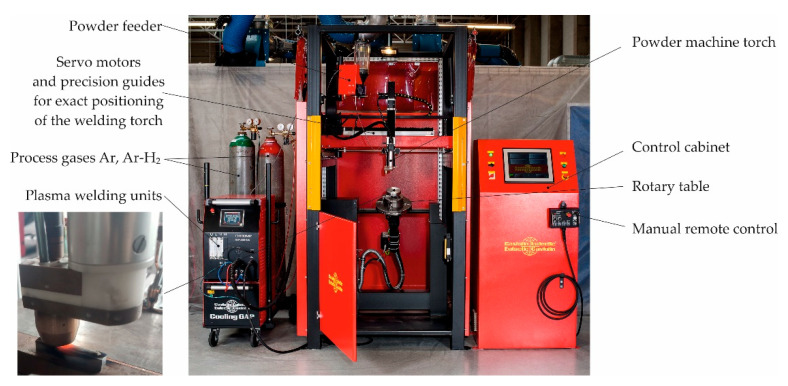
Gap automated unit robotic coating system.

**Figure 4 materials-14-06066-f004:**
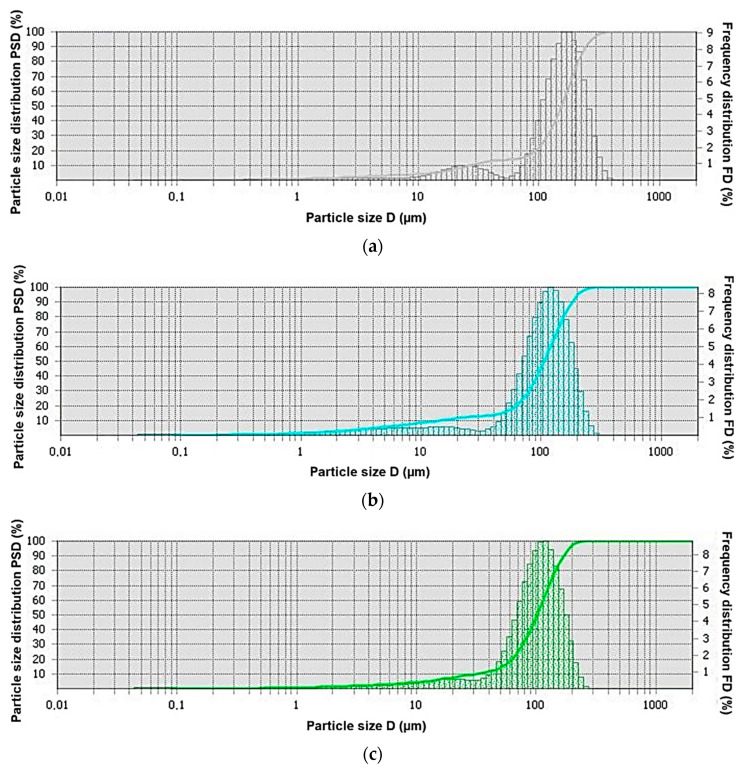
Graphs of cumulative percent passing versus the logarithmic particle size: (**a**) C1 filler material; (**b**) C2 filler material; (**c**) C3 filler material; (**d**) C4 filler material.

**Figure 5 materials-14-06066-f005:**
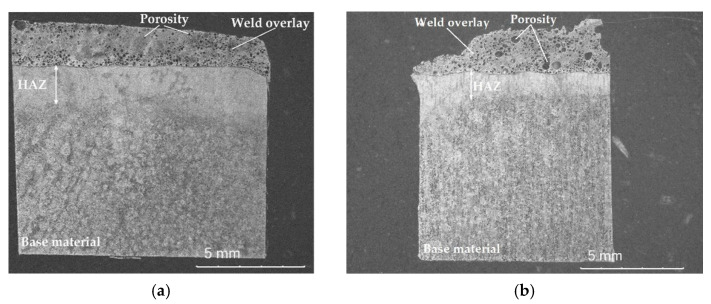
Exemplary macroscopic image of the longitudinal section of a composite surfacing weld: (**a**) C1 filler material; (**b**) C3 filler material.

**Figure 6 materials-14-06066-f006:**
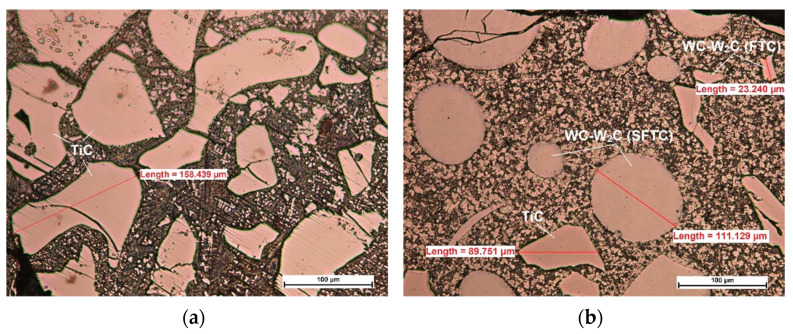
Light microscopy image of the structure of the composite layer padded using the PPTAW method with powder on the alloy matrix: (**a**) cobalt with a hard reinforcing phase in the form of crushed TiC and spherical PD-W particles (C1 filler material); (**b**) nickel with a hard reinforcing phase in the form of crushed TiC and spherical broken tungsten carbide WC-W2C particles (C2 filler material); (**c**) nickel with a hard reinforcing phase in the form of particles of WC-W_2_C spherical and broken tungsten carbide and spherical polycrystalline synthetic diamond (C3 filler material); (**d**) nickel with a hard reinforcing phase in particles of spherical tungsten carbide WC-W_2_C and titanium diboride TiB_2_ (C4 filler material); (**e**) microstructure of the parent material of AISI 4715 non-alloy structural steel.

**Figure 7 materials-14-06066-f007:**
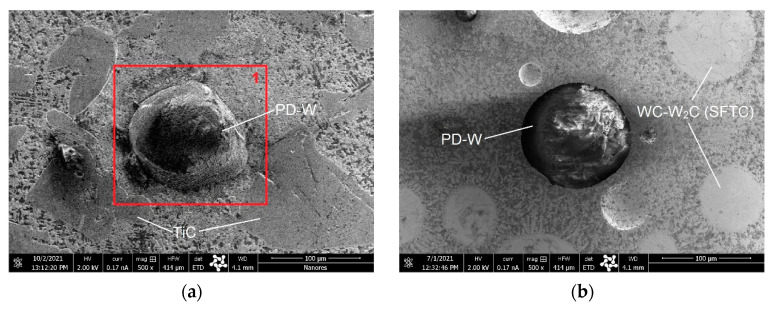
SEM image of the structure of selected composite layers padded using the PPTAW method with a powder on an alloy matrix: (**a**) cobalt with a hard reinforcing phase in the form of particles of crushed TiC and spherical polycrystalline synthetic diamond (C1 filler material); (**b**) nickel with a hard reinforcing phase in the form of particles of WC-W_2_C spherical and broken tungsten carbide and PD-W spherical polycrystalline synthetic diamond (C3 filler material).

**Figure 8 materials-14-06066-f008:**
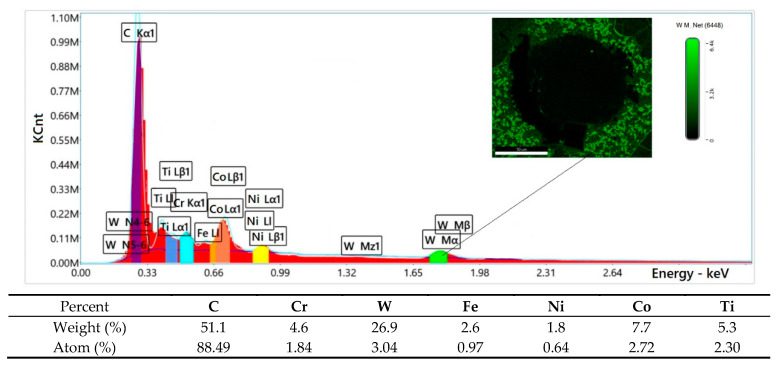
Diagram of scattered X-radiation energy with energy lines present in area 1 (Figure 7a) of the analyzed components (chemical elements).

**Figure 9 materials-14-06066-f009:**
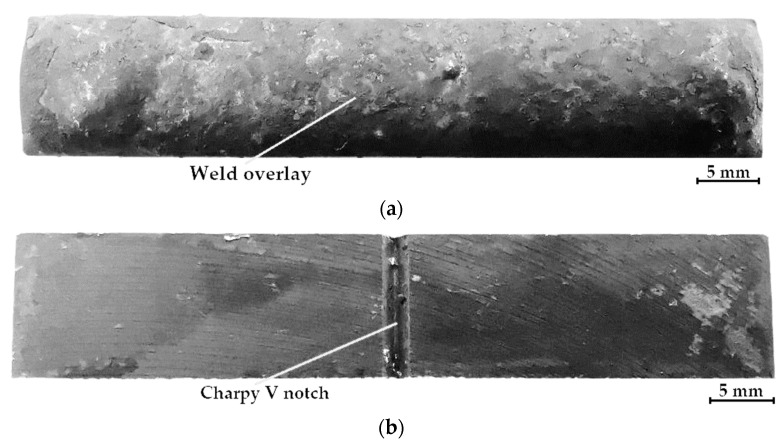
View of the sample before the Charpy impact test: (**a**) view from the weld overlay; (**b**) view from the Charpy V notch.

**Figure 10 materials-14-06066-f010:**
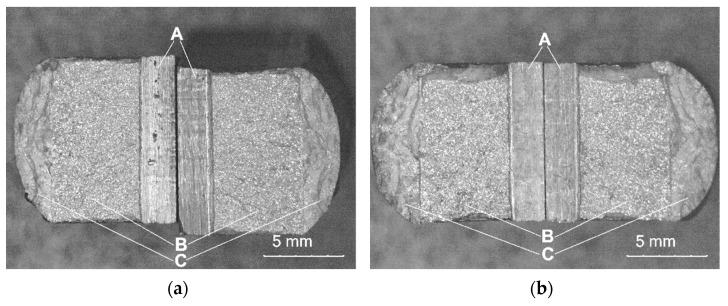
Fractures after impact tests at a temperature of +23 °C, A—notch area, B—area of the dynamic propagation zone of the fracture front in the native material, C—area of the dynamic propagation zone of the fracture front in the layer deposited with powder on a matrix of: (**a**) C1 filler material; (**b**) C2 filler material); (**c**) C3 filler material; (**d**) C4 filler material; (**e**) parent material.

**Figure 11 materials-14-06066-f011:**
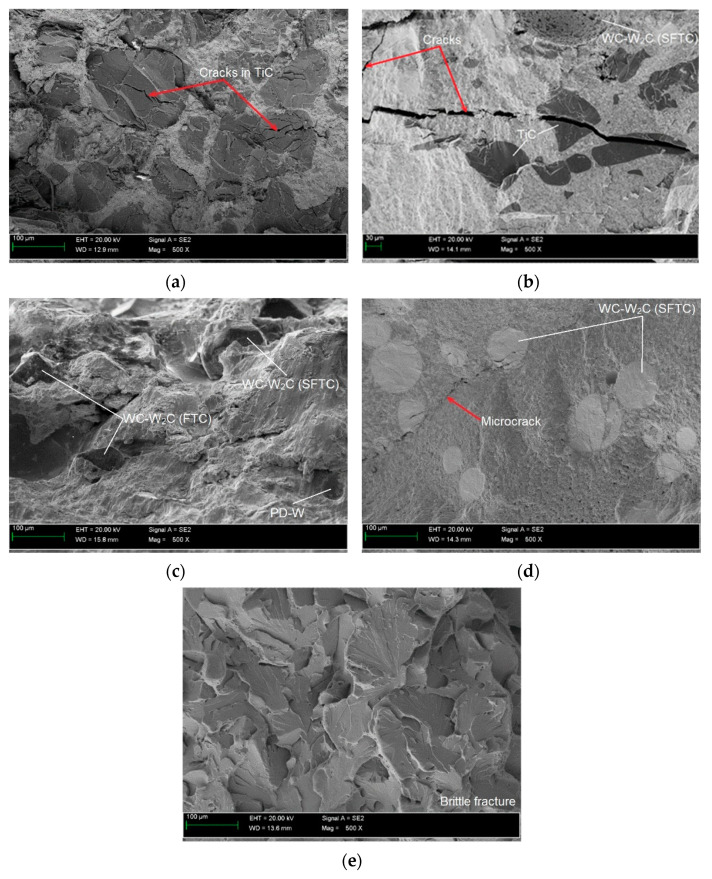
SEM images of the brittle fracture surface for samples: (**a**) C1; (**b**) C2; (**c**) C3; (**d**) C4; (**e**) base material.

**Figure 12 materials-14-06066-f012:**
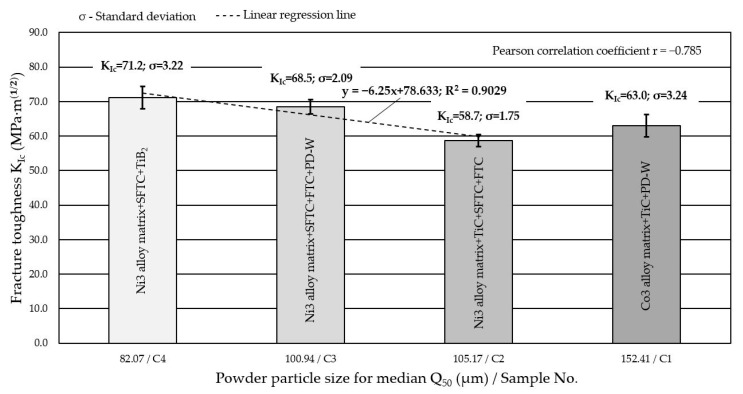
Influence of the particle size of the powder used for surfacing composite layers on the critical value of the stress intensity factor *K*_Ic_.

**Figure 13 materials-14-06066-f013:**
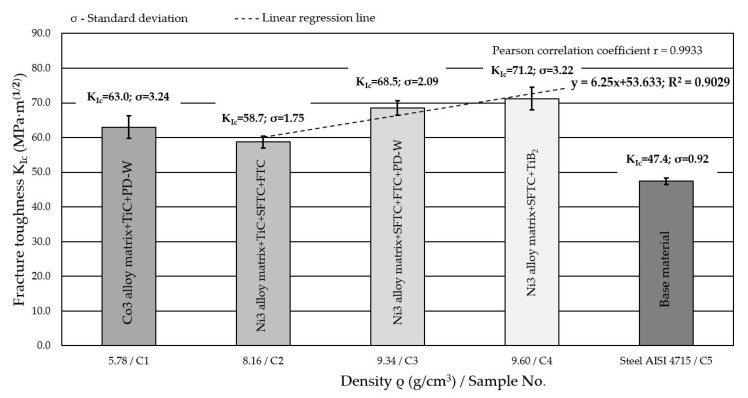
Influence of composite layer density on the critical value of the stress intensity factor *K*_Ic_.

**Figure 14 materials-14-06066-f014:**
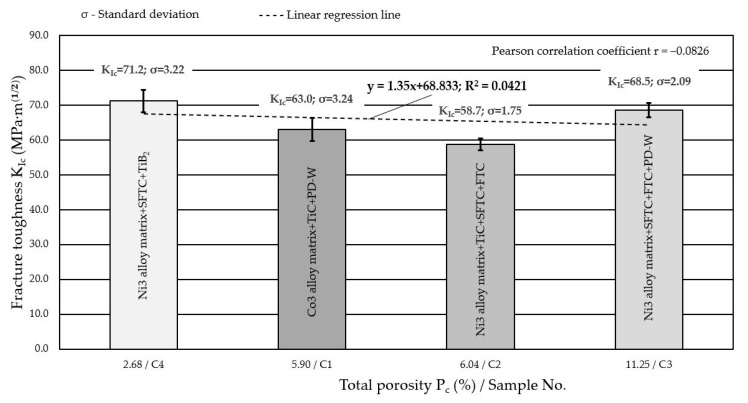
Influence of porosity of composite layers on the critical value of the stress intensity factor *K*_Ic_.

**Table 1 materials-14-06066-t001:** Properties of ceramic materials used to reinforce the metal matrix in surfaced composite layers.

Composite Reinforcement Properties	TiC	WC	TiB_2_	PD ^1^
Density, g/cm^3^	4.93	15.8	4.52	3.51
Decomposition temperature, °C	3065	2870	2980	1450
Thermal expansion coefficient at 298 K, K^−1^	7.74 × 10^−6^	6 × 10^−6^	3.5 × 10^−6^	1.3 × 10^−6^
Thermal conductivity at 298 K, W/(m·K)	50	90–110	96	550
Young’s modulus, GPa	410–510	550	510–570	1050
Compressive Strength, GPa	1.8	2–2.5	1.8	4.2
Flexural Strength, MPa	240–390	350	400–450	850
$\mathrm{Fracture}\;\mathrm{toughness}\;K_{\mathrm{Ic}},\;\mathrm{MPa}\cdot m^{\frac{1}{2}}$	4–5.5	4.5	4–6	8.5
Hardness, GPa	28–35	30	25–35	75
Other features	High chemical resistance, very high abrasion resistance, relatively good resistance to oxidation	High chemical resistance, very high abrasion resistance, weaker resistance to oxidation	Very high abrasion resistance, good chemical resistance, very good resistance to liquid metals, resistance to oxidation (in the air) to 1000 °C	Low friction coefficient, high strength, very good chemical resistance, biocompatibility, poor resistance to oxidation around 780 °C
Reference	[11]	[12]	[7]	[10]

^1^ Polycrystalline synthetic diamond made by high pressure and high temperature (HPHT) technology.

**Table 2 materials-14-06066-t002:** Examples of chemical compositions of filler materials used to protect the working surface of drilling tools.

Hard-Facing Alloy			Reference
Composition, wt.%		Carbide-to-Matrix Ratio	
Carbide Reinforcement	Matrix		
WC-CoCr (FTC)	Ni–17Cr–4Fe–4Si–3.5B–1C	70/30	[14]
WC-W_2_C (FTC)	Ni–0.2C–3.5–Si–3Fe–2.3B	60/40	[15]
WC-W_2_C (FTC)	Ni–7.5Cr-3Fe-3.5Si–1.5B-0.3C	55/45	[4]
WC-W_2_C (SFTC)	Ni–0.1C–3Si–3B–2Fe	60/40	[16]
WC-W_2_C (SFTC)	Ni–9.5Cr–3Fe–3Si–1.6B–0.6C	68/32	[4]
WC-W_2_C (FTC)	Co–27Mo–16.9Cr–3Si–0.5Fe–0.8Ni	35/65	[17]
WC-W_2_C (SFTC)	Ni– < 3Si– < 1B	40–90/60–10	[4]
TiB_2_	Ti–6Al–4V	50/50	[18]

**Table 3 materials-14-06066-t003:** Chemical composition of powders.

Filler Material	Chemical Composition of the Matrix, wt. (%)										Ceramic Reinforcement of the Matrix, wt. (%)				
	C	Si	Mn	Cr	B	Mo	W	Fe	Ni	Co	TiB_2_	TiC	SFTC	FTC	PD-W
C1	2.5–3	≤1	≤2	24–28	-	≤1	12–14	<5	≤3	Bal.	-	90	-	-	10
C2	≤0.05	<2.4	0.5	2.0	≤1.4	-	-	≤0.5	Bal.	-	-	60	30	10	-
C3	≤0.05	<2.4	0.5	2.0	≤1.4	-	-	≤0.5	Bal.	-	-	-	70	10	20
C4	≤0.05	<2.4	0.5	2.0	≤1.4	-	-	≤0.5	Bal.	-	10	-	90	-	-

Carbide-to-matrix ratio: 60/40 (acc. to %). Bulk density of the powder determined by pycnometry: C1 = 7.47 g/cm^3^, C2 = 11.03 g/cm^3^, C3 = 11.64 g/cm^3^, C4 = 13.80 g/cm^3^.

**Table 4 materials-14-06066-t004:** Chemical composition of low-alloy structural steel AISI 4715 according to the manufacturer’s data (TimkenSteel Ltd., Canton, OH, USA).

Chemical Composition, wt.%								
C	Mn	S	P	Si	Cr	Mo	Ni	Fe
0.12–0.18	0.65–0.95	≤0.015	≤0.015	0.15–0.35	0.40–0.70	0.45–0.60	0.65–1.00	Bal.

Notes: the average density of steel at a temperature of 25 °C, ρ = 7.865 g/cm^3^.

**Table 5 materials-14-06066-t005:** Cladding parameters of the plasma transfer arc welding (PTAW) metal deposition of the surface layers on AISI 4715 steel.

Process Parameters	Value of Parameter
Current, *I* (A)	80
Voltage, *U* (V)	25
Travel speed, *S* (mm/s)	2.7
Powder feed rate, *q* (g/min)	18
Heat input, *E*_u_ ^1^ (J/mm)	444

Notes: Argon 5.0 (99.999%) acc. ISO 14175—I1: 2009 [27] was used as the plasma gas (flow rate = 1.6 L/min), argon/hydrogen 5% H_2_, Ar (welding mixture ISO 14175-R1-ArH-5) was used as shielding gas (flow rate = 12 L/min) and carrier gas (flow rate = 4 L/min), ^1^ calculated acc. to the formula: *E*_u_ = *k*∙(*U* × *I*)/*S*. The thermal efficiency coefficient for plasma-transferred arc *k* = 0.6 was used.

**Table 6 materials-14-06066-t006:** Statistical mean values of the particle size distribution of the powders used for surfacing (PPTAW) of composite layers.

Average Statistical Value	Sample Determination			
	C1	C2	C3	C4
Quantile *Q*_10_ (µm)	26.60	14.17	31.22	5.13
Median *Q*_50_ (µm)	152.41	105.17	100.94	82.07
Quantile *Q*_90_ (µm)	247.55	180.10	167.32	162.88

**Table 7 materials-14-06066-t007:** Density-related measurement results and calculations of the porosity of the surface layers deposited (PPTAW) on structural low-alloy AISI 4715 steel.

Physical Quantity	Average			
	C1	C2	C3	C4
Density *ρ* (g/cm^3^)	5.7785	8.1582	9.3425	9.6013
Standard deviation *σ_ρ_*	0.2117	0.6267	0.3150	0.2349
Absorbability *A* (%)	1.0219	0.6368	1.3856	0.1315
Open porosity *P_o_* (%)	5.6467	0.6368	11.2421	1.2346
Closed porosity *P_c_* (%)	0.2526	5.7292	0.0082	1.4318
Apparent density *ρ_a_* (g/cm^3^)	5.4787	7.7335	8.9729	9.3400
Total porosity *P_c_* (%)	5.8993	6.0366	11.2503	2.6778

**Table 8 materials-14-06066-t008:** Fracture toughness *K*_Ic_ and critical value of crack opening *CTOD* calculated based on the value of the average impact work of breaking *KV* at 23 °C of the AISI 4715 steel parent material and steel samples with wear-resistant composite surfacing welds.

Sample No. (Composite Weld Overlay)	*KV* (J) ^1^	Standard Deviation σ (J)	$K_{\mathrm{Ic}\;\mathrm{Equation}\;(3)}\;(\mathrm{MPa}\cdot m^{\frac{1}{2}})$	*CTOD* _Equation (5)_ (mm)
C1 (Co3 + TiC + PD-W)	18.9	1.9	63.0	0.045
C2 (Ni3 + TiC + SFTC + FTC)	16.4	1.0	58.7	0.039
C3 (Ni3 + SFTC + FTC + PD-W)	22.3	1.4	68.5	0.054
C4 (Ni3 + SFTC + TiB_2_)	24.1	2.2	71.2	0.058
C5 (Steel AISI 4715)	10.7	0.4	47.4	0.026

^1^ Average value of three test results.

## Data Availability

The data are not publicly available due to the initiation of a patent procedure (No. P435997).

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
