# Peer review of "Matrix Composite Coatings Deposited on AISI 4715 Steel by Powder Plasma-Transferred Arc Welding. Part 3. Comparison of the Brittle Fracture Resistance of Wear-Resistant Composite Layers Surfaced Using the PPTAW Method"

_materials, 2021, doi:10.3390/ma14206066_

Round 1
Reviewer 1 Report
The abstract is too long with unnecessary information. Please focus on the key points and make it concise.
English needs significant improvement throughout the manuscript.
Research gap and novelty must be clearly highlighted at the end of Introduction.
Please explain how did determine these parameters? are the optimum?
your figure captions are too long. Make them concise. for example, Figure 2 caption: define C1 TO C4 within the text and remove the description from all figure captions but keep (a) C1 binder (a) C2 binder ....etc
Make the conclusions section concise
Figure 11 caption. missing information
Author Response
Dear Reviewer 1,
I would like to thank you for your valuable information concerning the article entitled “Comparison of the brittle fracture resistance of wear-resistant composite layers surfaced using the PPTAW method”
We humbly accept your justified remarks, which undoubtedly improved the article. The publication has been corrected accordingly. We are fully aware of certain scientific shortcomings. We do intend to eliminate them in future. All the corrections have been made in the attached manuscript. The publication has been modified following all the instructions contained in your review:
- According to your suggestion, the article has been submitted for linguistic proofreading. The proofreading work was done by a specialized English translator (certificate attached).
- I agree entirely, the abstract of the article is too long and contains unnecessary information. I made the appropriate adjustments.
- As your suggestion at the end of the article introduction, the research gap and novelty are clearly highlighted.
- According to your request, I explained how the optimal silicon and boron contents of the filler material were determined.
- You are absolutely right. The captions under the figures are too long. I have shortened them according to your directions.
- Thank you for pointing out that the caption in Figure 11 was incorrect. The bug has been fixed.
- As you suggested, the conclusions have been reformatted and shortened. I hope you will be satisfied with this.
Let me emphasize that your comments have undoubtedly helped me improve the article. I will do my best to eliminate any scientific shortcomings in the future. I tried to address all Reviewers’ remarks (sometimes contradictory). I do hope that you understand the importance of this publication for me. Should you have any questions, please do not hesitate to contact me (artur.czuprynski@polsl.pl).
Thank you very much for your help.
Yours faithfully,
Artur Czupryński

Reviewer 2 Report
Dear authors, you have presented a very interesting and important work based on a wide range of experimental data. However, there are a number of comments that you should pay attention to.
1. The text of the manuscript is difficult to read, especially in the abstract, which should be simple and understandable to any reader. Complex sentences are often stretched over 3 or more lines and may be unclear to readers. Please try to make corrections where possible, maybe with the advice of a professional.
2. The results of the paper look like just a presentation of figures. It is necessary to describe the results in more detail. This will better prepare the reader for the discussion and avoid the repeated need to go back to previous sections when reading the discussion.
3. In the discussion and conclusions you mention the temperature gradient in the PPTAW process. Have you evaluated this gradient in your work?
4. There are typos in the text (e.g., line 396). Carefully read the manuscript and make corrections.
Author Response
Dear Reviewer 2,
I would like to thank you for your valuable information concerning the article entitled “Comparison of the brittle fracture resistance of wear-resistant composite layers surfaced using the PPTAW method”
We humbly accept your justified remarks, which undoubtedly improved the article. The publication has been corrected accordingly. We are fully aware of certain scientific shortcomings. We do intend to eliminate them in future. All the corrections have been made in the attached manuscript. The publication has been modified following all the instructions contained in your review:
- According to your suggestion, the article has been submitted for linguistic proofreading. The proofreading work was done by a specialized English translator (certificate attached).
- The abstract has been corrected according to the requirements of the journal.
- In line with your suggestions, the research results have been described and further commented on in the discussion.
- You're right. In the article, I refer to a large temperature gradient without presenting specific research results. However, in the case at hand, the process was carried out with a relatively low heat source power and thermal efficiency coefficient of the process and a high welding speed. These welding conditions result in a large temperature gradient. Details of these studies will be published shortly. The calculation results of the temperature field during single-beads PPTAW cladding of the AISI 4715 steel plate will be presented in the next paper (Authors' article: Winczek J., Kik T., Czupryński A.: "Numerical simulation of a temperature field during single-beads surface welding" will be submitted to Journal of Applied Mathematics and Computational Mechanics. Numerical simulations were performed using the SysweldR program. Two of Goldak’s heat source models were chosen for calculating the temperature field for padding weld.
- All typos have been corrected in the text of the article.
We hope that's what you meant. Let me emphasize that your comments have undoubtedly helped me improve the article. I will do my best to eliminate any scientific shortcomings in the future. I tried to address all Reviewers’ remarks (sometimes contradictory). I do hope that you understand the importance of this publication for me. Should you have any questions, please do not hesitate to contact me (artur.czuprynski@polsl.pl).
Thank you very much for your help.
Yours faithfully,
Artur Czupryński

Reviewer 3 Report
Please find my comments in the attached pdf.

Author Response
Dear Reviewer 3,
I would like to thank you for your valuable information concerning the article entitled “Comparison of the brittle fracture resistance of wear-resistant composite layers surfaced using the PPTAW method”
We humbly accept your justified remarks, which undoubtedly improved the article. The publication has been corrected accordingly. We are fully aware of certain scientific shortcomings. We do intend to eliminate them in future. All the corrections have been made in the attached manuscript. The publication has been modified following all the instructions contained in your review:
- At the outset, I would like to inform you that the text of the article has been checked by a linguist (the certificate is attached).
- Following your advice, I removed unnecessary abbreviations repeated in parentheses.
- According to your suggestion, the article has been submitted for linguistic proofreading. The proofreading work was done by a specialized English translator (certificate attached).
- In line with your suggestions, the research results have been described and further commented on in the discussion.
- You're right. I admit that I misused the word weight instead of mass. The term constant mass is used to define when a sample is dry.
- I have added a description explaining the features observed in macrostructures.
- As you asked, I added the values of the standard deviation in table 7. I only gave the value of the standard deviation for density. Will it be okay? I hope you will be satisfied.
- You are right. The enlargements of the images in Figure 7 were different. I corrected this error. It looks a lot better now. Thank you.
- Thank you for pointing out that the caption in Figure 11 was incorrect. The bug has been fixed.
We hope that's what you meant. Let me emphasize that your comments have undoubtedly helped me improve the article. I will do my best to eliminate any scientific shortcomings in the future. I tried to address all Reviewers’ remarks (sometimes contradictory). I do hope that you understand the importance of this publication for me. Should you have any questions, please do not hesitate to contact me (artur.czuprynski@polsl.pl).
Thank you very much for your help.
Yours faithfully,
Artur Czupryński

Round 2
Reviewer 1 Report
acceptable in present form